# Air Pollution Associates with Cancer Incidences in Poland

**Norbert Tuśnio [1],*[ID], Jakub Fichna [2], Przemysław Nowakowski [3][ID] and Piotr Tofiło [4][ID]**

1   Faculty of Safety Engineering and Civil Protection, The Main School of Fire Service, J. Slowackiego 52/54, 01-629 Warsaw, Poland
2   Department of Biochemistry, Medical University of Lodz, 92-215 Lodz, Poland; jakub.fichna@umed.lodz.pl
3   Independent Researcher, 85-113 Lodz, Poland; nowakowski.przemyslaw@gmail.com
4   Institute of Safety Engineering, The Main School of Fire Service, J. Slowackiego 52/54, 01-629 Warsaw, Poland; ptofilo@sgsp.edu.pl
*   Correspondence: ntusnio@sgsp.edu.pl; Tel.: +48-602-896-982

**Abstract:** In many countries around the world (including the United States, Canada, and Spain), research is being conducted into the impact of air pollution on the formation of various types of cancer. For a long time it was thought that the inhalation of pollutants could lead to lung diseases. Now the effects of air pollutants on tumors in the airways, kidneys, bladder, breast, and colon have been investigated and are better understood. It is now known that particulates in air pollution can cross the blood–brain barrier and also reach the placenta. The aim of this study was to find a possible relationship between the emission of pollutants into the atmosphere and the formation of specific types of tumors in the Polish population. Two databases available on the Internet were used in the analysis: the bank of measurement data on air quality in Poland (the repository of Environmental Protection Inspection) and cancer statistics. The pollution measurement data for the years 2000–2016 were taken from the Chief Inspectorate for Environmental Protection website, a database with results from 264 stations located in Poland for 13 types of gases and atmospheric pollutants. Statistical data on cancer C00–D09 (according to the *International Statistical Classification of Diseases and Related Health Problems*, 10th Revision (ICD-10)) in the Polish population in the years 1999–2015 were retrieved from onkologia.org.pl. A novel code was constructed, allowing the downloading of statistics from the databases, examination of their correlation, and selection of the best model of regression through machine learning. The results of the analyses indicate a high correlation of air pollution with the incidence of selected types of cancer. Particularly noteworthy is the observed effect of $NO_x$ on the incidence of small and large intestine cancers in the Masovia and West Pomerania provinces. The other gases and pollutants with the most significant impact on the incidence of gastrointestinal cancer have also been identified. Based on statistical analysis, we found a correlation between air pollution and tumor incidence in individual provinces, as well as an influence of the emission of nitrogen oxides on the cancer incidence rate.

**Keywords:** air pollution; environmental pollution; pollution measurements; neoplasms epidemiology; Poland

## 1. Introduction

Cancer is a group of diseases involving abnormal cell growth with the potential to invade or spread to other parts of the body. Humans are affected by over 100 types of cancers. Exposure to ionizing radiation and air pollution are listed among other carcinogenic factors. Air pollutants include oxides of sulfur, nitrogen and carbon, as well as benzene and smog-forming particles, namely PM2.5

(in which PM stands for particulate matter)—atmospheric aerosols (suspended dust) with a diameter of no more than 2.5 μm, which according to the World Health Organization are the most harmful to human health from other atmospheric pollutants, and PM10—a mixture of suspended particles with a diameter of no more than 10 μm. The composition may additionally include such toxic substances as, for example, benzopyrene, dioxins, and furans.

Air pollution occurs when harmful or excessive quantities of substances, including gases, particulates, and biological molecules, are introduced into the Earth's atmosphere. It may cause diseases, allergies and even death in the most severe cases [1].

For many years, studies have been conducted to examine the impact of air pollution on the risk and incidence of cancer. Canadian researchers investigated the impact of sulfur dioxide ($SO_2$) air pollution on mortality in breast and colon cancer patients in 20 cities in their country. It was assumed that $SO_2$ absorbs ultraviolet light in the region of the spectrum which is most active in forming vitamin D in the skin. Since vitamin D plays a role in reducing the risk of colon and breast cancer, high concentrations of the pollutant (acid haze) may lead to its deficiencies in exposed populations [2]. Indeed, statistically significant positive associations were found between air pollution and age-adjusted mortality rates for colon cancer in women and men, and breast cancer in women.

Researchers from Spain (Barcelona Institute of Global Health) and the United States (American Cancer Society) carried out a large-scale epidemiological study that linked certain air pollutants to kidney, bladder, and colon diseases. The study involved over 600,000 adults in the US who were followed up for 22 years (from 1982 to 2004). A team of researchers investigated a possible association of deaths from cancer in 29 places in the country with a long-term population exposure to three types of pollutants: PM2.5, nitrogen dioxide ($NO_2$), and ozone ($O_3$). Research has shown that PM2.5 was associated with mortality from kidney and bladder cancer, and exposure to $NO_2$ was associated with colorectal cancer death [3].

In 2009, the results of the long-term studies on geographical variation in cancer mortality rates were reported. Researchers from the Nutrition and Health Research Center, located in the United States, proved that there is a connection between atmospheric air pollution from fossil fuel combustion with the risk of cancer development in the digestive tract, elimination organs (esophageal and bladder), and female reproductive organs [4].

Studies carried out at the National Statistics Institute in Spain have allowed for the designation of industries that contribute to the emission of the most dangerous compounds into the atmosphere. These include mining, paper and wood production plants, the food industry, metal production and processing, and ceramics. Further studies were then conducted to verify whether the proximity of such industries that emit pollutants into the air could be an added risk factor for colorectal cancer mortality [5]. A summary of the past literature (only relevant studies) is presented in Table 1.

Poland belongs to the top European Union countries when it comes to air pollution. The very poor air quality that the inhabitants of many regions of Poland inhale should be viewed not only in terms of environmental degradation, but also as a huge example of neglected development for the country. The most polluted regions, based on a World Health Organization ranking of the EU cities with the most polluted air, are Lower Silesia, Silesia, and Lesser Poland.

As of today, only the tests carried out by the Environmental Protection Inspectorate provide the most reliable information on the state of the atmospheric air in Poland, as they are subject to rules of control and measurement quality. Of note, characteristics and assessments of the quality of the environment can also be made with the use of drones. This method is an alternative to expensive photogrammetric and time-consuming field measurements. Moreover, it is characterized by high mobility (flights below the cloud level, high time resolution, and data registration even for small areas). This method applies to the study of air quality, as well as water analysis.

**Table 1.** A table summarizing the past literature including types of pollutants studied, levels of pollutants, study designs and target population as compared to the data in the current study. PM: particulate matter.

| | Types of Pollutants | Levels of Pollutants | Study Design | Target Population | Reference |
|---|---|---|---|---|---|
| 1. | $SO_2$ | high concentrations (acid haze) | age-adjusted breast and colon cancer mortality rates were examined | 20 cities in Canada | [2] |
| 2. | PM2.5, $NO_2$, $O_3$ | particulate matter concentrations were estimated for the participant residence | mortality due to kidney, bladder and colorectal cancers were observed | 623,048 participants | [3] |
| 3. | acid deposition, carbonaceous aerosols, polycyclic aromatic hydrocarbons | not only ambient pollution but occupational exposure, smoking, and polycyclic aromatic hydrocarbons (PAH) | age-adjusted, gender-specific and race-specific cancer mortality rates were obtained (20 groups of cancer) | population of the United States (data up to 1994) | [4] |
| 4. | pollution emitted by various industries | population exposure to industrial pollution was estimated by reference to the distance from town centroids to industrial facilities | the relative risks (RR) of towns situated at a short distance from industrial installations were estimated (colorectal cancer mortality) | 8098 towns in Spain | [5] |
| 5. | $SO_2$, $NO_2$, $NO_x$, CO, $O_3$, $C_6H_6$, PM10, PM2.5, Pb(PM10), As(PM10), Cd(PM10), Ni(PM10), BaP(PM10) | measurement data of the Chief Inspectorate of Environmental Protection (Poland) | correlation of air pollution with selected cancer cases by 16 provinces | 2,153,652 cancer cases in Poland (2000–2015) | our study |

In Poland, the earliest studies on the correlation between air pollution and cancer risk concerned the impact of environmental pollution on the incidence of malignant neoplasms of the upper respiratory tract, mainly in the region of Silesia [6,7]. Further studies extended the research area to all 16 provinces, distinguishing the type of gas or pollution occurring in a given area of the country. Concurrently, the problem of cancer in Poland has been described in [8,9], and the statistics of cases from 1999–2015 are presented in Figure 1.

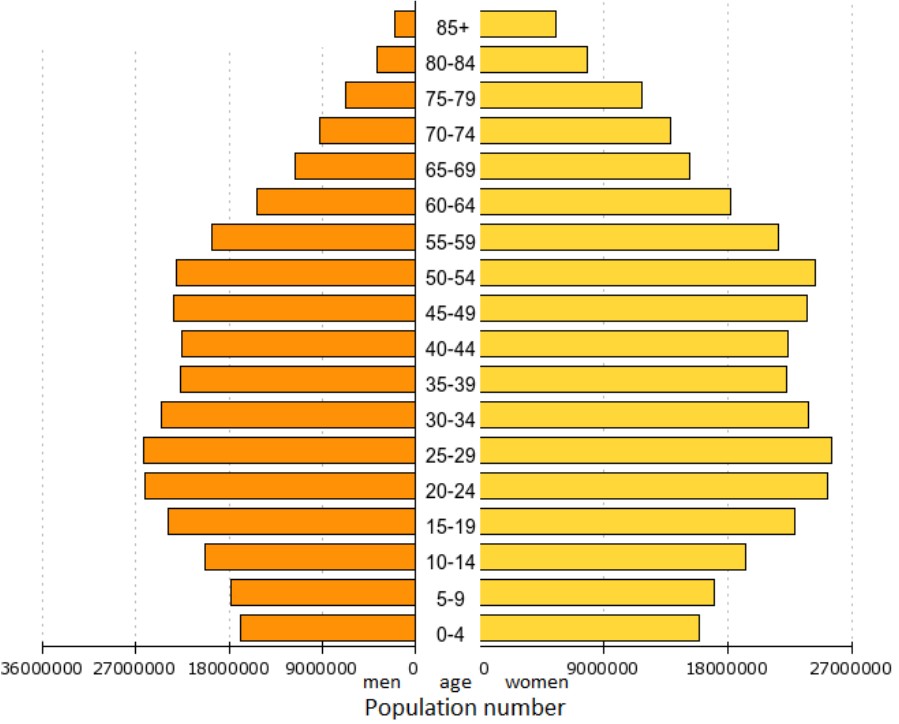

**Figure 1.** The number of cancer cases in Poland in the years 1999–2015 (age pyramid). Source: National Cancer Registry [10].

The aim of our study was to indicate which types of cancer are characterized by the highest correlation with the emission of selected types of air pollutants. As a consequence of individual stages of research, the most important gases and pollutants influencing the selected type of cancers (malignant neoplasm of the bronchus and lung, and both the small and large intestine) were determined. This choice was dictated by the national disease statistics—these were the most common cancers.

Scientists from Taiwan conducted a very similar study to ours [11]. Annual mean concentrations of each air pollutant were determined at 75 air quality monitoring stations, and the concentrations were extrapolated for 349 local Taiwanese administrative areas. In total, 70 correlation coefficients between cancer incidence rates and various air pollutants were calculated. A significantly positive correlation was observed between the level of PM2.5 and the cancer incidence rate after multiple testing corrections.

## 2. Methods

Two databases were used in this work: a database with the results of pollutant measurements (the repository of Environmental Protection Inspection) [12] and statistics on the formation of tumors [10]. The former contains measurements of gases and pollutants contained in the air carried out in Poland in the years 2000–2016. The list of measured substances includes $SO_2$, $NO_2$, $NO_x$, CO, $O_3$, $C_6H_6$, PM10, PM2.5, Pb(PM10), As(PM10), Cd(PM10), Ni(PM10), and BaP(PM10).

The Environmental Protection Inspection tests the PM10 and PM2.5 emission and content in the air using two complementary methods: the gravimetric (reference) method (approx. 250 locations)

and the automatic (approx. 180 locations) method. The data is read out every hour, and it is verified in a 4-stage system: ongoing, periodic, annual, and national verification. Two types of devices are used:

1. Dust collectors operating on the basis of reference methodologies. Collectors are produced by the companies Comde Derenda GmbH, MCZ GmbH, and Sven Leckel.
2. Meters operating in online measurement mode, according to the methodology equivalent to the reference method. These meters are manufactured by the companies Envea, Grimm Aerosol Technik, PALAS GmbH, and Thermo Fisher Scientific (US).

For gaseous pollutants (CO, $SO_2$, $NO$-$NO_2$-$NO_x$, $O_3$, and BTEX—volatile aromatic hydrocarbons), analyzers from companies such as Teledyne API, Thermo Fisher Scientific, Envea, Horiba, Synspec B.V., AMA Instruments GmbH, and Chromatotec are used.

The second database covers epidemiological data on the formation of cancer in Poland in the years 1999–2015, with the division into provinces and counties, types of disease according to the *International Statistical Classification of Diseases and Related Health Problems*, 10th Revision (ICD-10) and patient gender. It should be noted that in Poland there is an obligation to submit a Malignant Cancer Notification Card to doctors (data came from this source). The data available in the two databases are compared in Table 2.

**Table 2.** Data available from the website powietrze.gios.gov.pl and onkologia.org.pl.

| Parameter | powietrze.gios.gov.pl | onkologia.org.pl |
|---|---|---|
| Years | 17 (2000–2016) | 17 (1999–2015) |
| Number of provinces | 16 | 16 |
| Number of | pollutants: 13 | neoplasms: 101 (C00-D09) |
| Method of obtaining data | Excel file | POST request * |

* POST is a request method supported by the HTTP protocol used by the World Wide Web.

We then calculated the correlation between the read value of concentrations of dangerous gases (on an annual scale) and the number of cancer cases (the cancer incidence rate, determined by the number of cases or deaths per 100,000 people tested). To perform the statistical analysis, a Python code [13] was used to estimate the Pearson correlation coefficient and the random forest regression algorithm results. The Pearson product-moment correlation coefficient is received using the NumPy package, in which the main parameters are two arrays containing multiple variables and observations (each row represents a variable, and each column a single observation of all those variables). A random forest regression algorithm is taken from Scikit-learn, which is an open source machine learning library that supports supervised and unsupervised learning. In this procedure each tree in the ensemble is built from a sample drawn with replacement from the training set. The source code used for analysis is posted on the Github repository (https://github.com/ntusnio/CV/blob/master/Projekt%20rak.ipynb).

The Pearson correlation coefficient is a measure of the linear correlation between two variables. It has a value between +1 and −1, where 1 is total positive linear correlation, 0 is no linear correlation, and −1 is total negative linear correlation.

The random forest algorithm [14] is a flexible, easy to use machine learning algorithm that produces, even without hyper-parameter tuning, an accurate result most of the time. It is also one of the most used algorithms because of its simplicity and the fact that it can be employed for both classification and regression tasks. The second method was used to assess the qualitative impact of individual types of hazardous gases on the statistics of the formation of small and large intestine cancers. The random forest algorithm was introduced in 1995 and for research purposes its results were verified on the basis of calculating the average accuracy of 1000 calculations on a separate part of the test data. It turned out that it gives better results than the XGBoost algorithm and the Lasso method. The random forest algorithm computes qualitative effects based on a feature importance score that indicates how useful or valuable each feature was in the construction of the boosted decision trees

within the model (the more an attribute is used to make key decisions with decision trees, the higher its relative importance).

The latency period for selected neoplasms was not included in the analyses. For example, lung cancer can usually occur 10–40 years from the onset of exposure. In addition, smoking (active or passive) and occupational exposure to inhalation of asbestos increase the risk of lung cancer development, and the research confirmed the synergistic effect of both of these factors. In order to prevent cancer, it is also necessary to eliminate additional factors contributing to the occurrence of cancer or mesothelioma, i.e., avoiding exposure to aromatic hydrocarbons.

The reason the latency period was not included in the analyses was because there was a wide range of neoplasms (over 100 types, according to the International Classification of Diseases), each of which is characterized by a different value, but also related to individual features. The consequences of adopting a zero latency period are similar to assuming an inappropriate value for the carcinogenesis period. As a result, the level of pollutants identified in the air in a given year will be correlated with the number of cancer cases in the year with an incorrectly selected delay. Due to the complexity and diversity of the process of changes taking place in the body's cells, leading to the formation of cancer, as well as the fact that carcinogenesis is a long-term process (the average period of development of a tumor with a diameter of 1 cm is about 5 years, although it depends on the type of tumor and tissue) it was decided to follow an approach that does not take into account the delay in cancer formation. Thus, following this assumption, the incidence of a given type of cancer was examined in a geographical area where a given type of air pollution occurs.

The analyses did not take into account any statistical methods other than correlations, and the research was limited to comparing the content of the two available databases. The limitations associated with such an approach resulted in not taking into account other factors leading to the formation of cancer, which include the presence of carcinogens, such as physical carcinogens (e.g., ultraviolet radiation), chemical carcinogens (alcohol and tobacco addiction, occupational exposure), or biological carcinogens (some viruses).

It should be added that many studies on the influence of air pollution on cancer risk have been conducted in Poland, and their results are presented, for example, in [15–19].

## 3. Results

First, the possible correlation between air pollution and cancer formation for all provinces in Poland was verified. For each pollutant, the best correlated cancer type was identified (Figure 2).

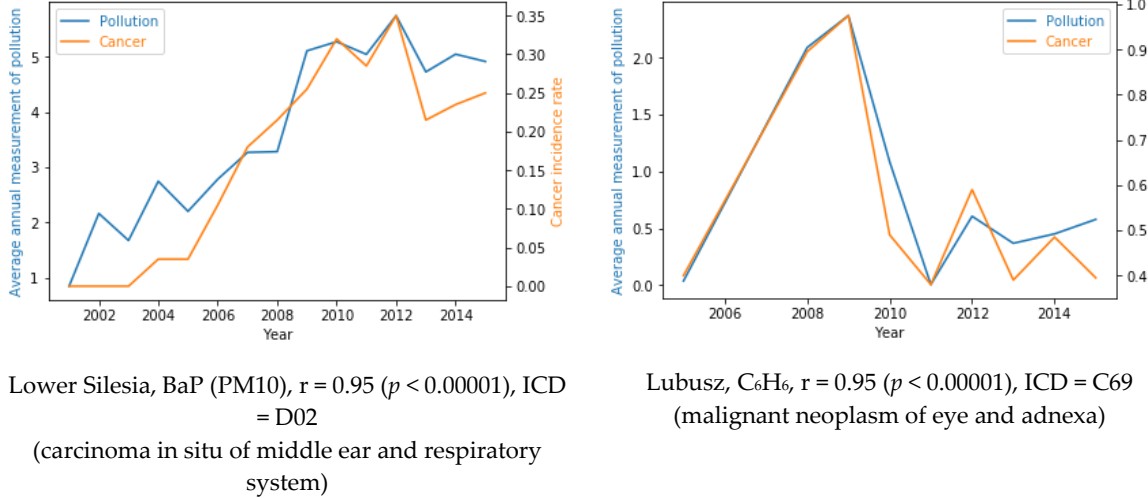

Lower Silesia, BaP (PM10), r = 0.95 ($p < 0.00001$), ICD = D02
(carcinoma in situ of middle ear and respiratory system)

Lubusz, $C_6H_6$, r = 0.95 ($p < 0.00001$), ICD = C69 (malignant neoplasm of eye and adnexa)

**Figure 2.** *Cont.*

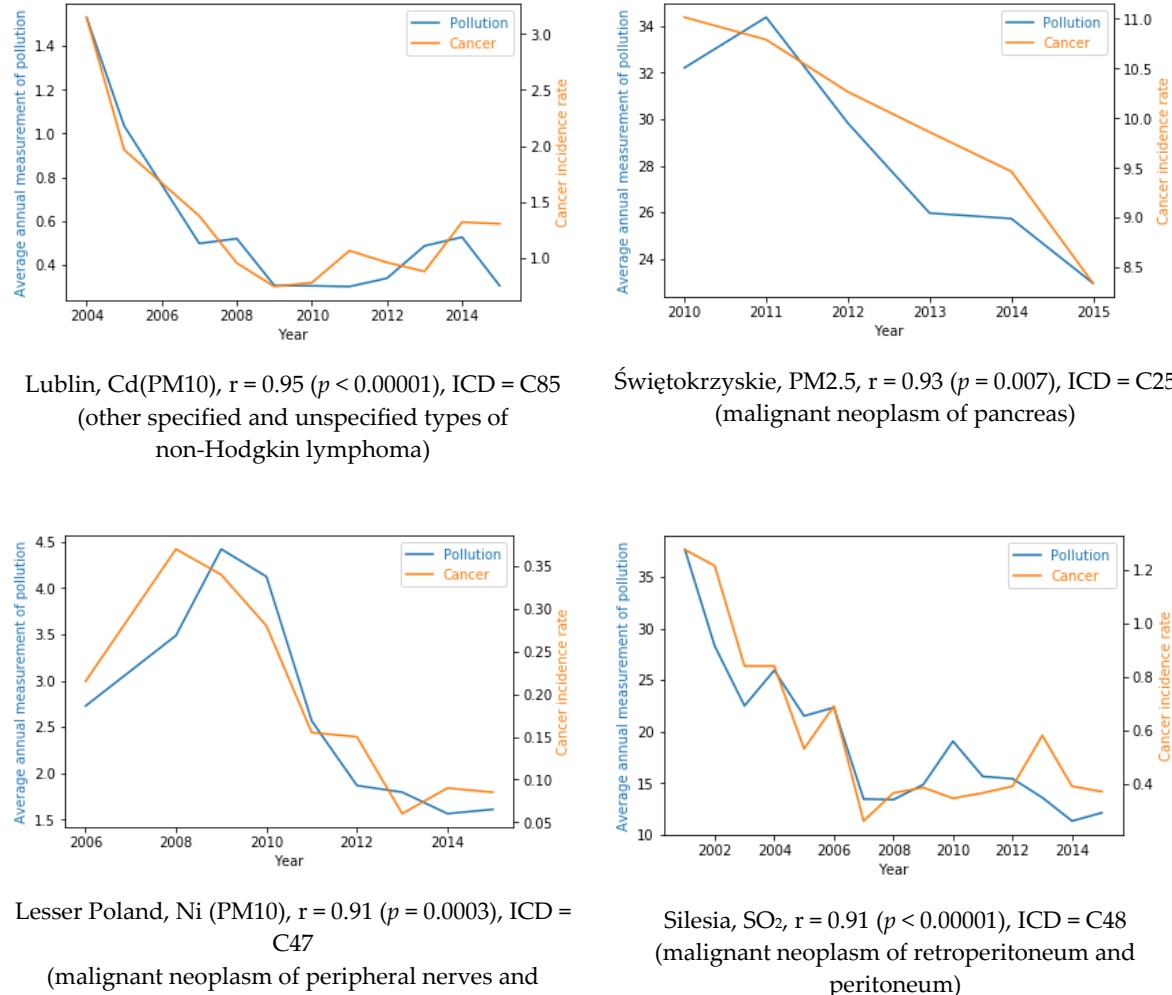

Lublin, Cd(PM10), r = 0.95 (*p* < 0.00001), ICD = C85 (other specified and unspecified types of non-Hodgkin lymphoma)

Świętokrzyskie, PM2.5, r = 0.93 (*p* = 0.007), ICD = C25 (malignant neoplasm of pancreas)

Lesser Poland, Ni (PM10), r = 0.91 (*p* = 0.0003), ICD = C47 (malignant neoplasm of peripheral nerves and autonomic nervous system)

Silesia, $SO_2$, r = 0.91 (*p* < 0.00001), ICD = C48 (malignant neoplasm of retroperitoneum and peritoneum)

**Figure 2.** The highest correlations between average air pollution ($\mu g/m^3$) and cancer formation in selected provinces of Poland, where r stands for the Pearson correlation coefficient and ICD is the International Classification of Diseases code of the disease.

Next, the most important pollutant was selected, based on the summation of the correlation values (r) for all cancer cases (C00–D09). Calculations were made for the whole country (for individual provinces these were not carried out). The results of this comparison are shown in Figure 3. The best correlation with the incidence of cancer was noted for the emission of nitrogen oxides.

Based on the results mentioned above, we focused our further analyses on nitrogen oxides and examined the correlation of their presence in the air at the monitoring sites with the formation of various types of tumors in each province. Table 3 shows the correlations in given provinces with the type of cancer for $NO_2$ and $NO_x$ emissions.

The choice of the type of cancer that were examined in detail was guided by the ratio of deaths to malignant neoplasms in Poland in the period available in the database (Figure 4), as well as the trends of the highest increases in disease.

Of note, our findings showed that the highest rate of increase in the number of cases in Poland is related to colorectal cancer, yet the literature suggests that cancers of the bronchus and lung cause the most deaths among men and women in the country [20].

Detailed results for intestinal cancer in correlation with air pollution are presented in Figure 5, in which C17 refers to the small intestine and C18 to the large intestine.

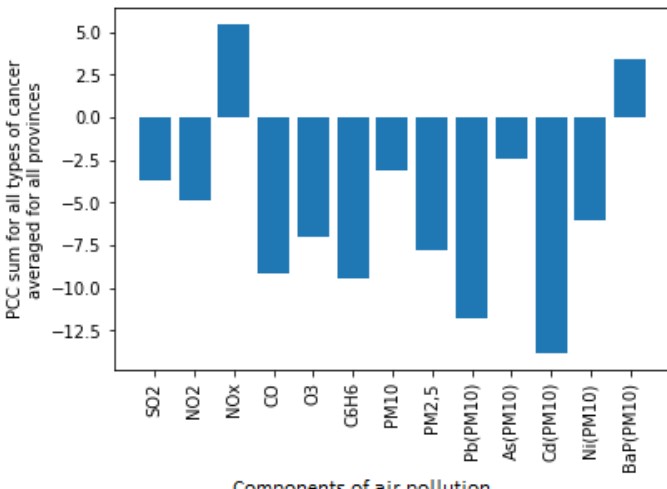

**Figure 3.** The level of correlation of a given pollutant with cancer incidents. PCC stands for the sum of the Pearson correlation coefficients (r).

**Table 3.** The highest correlations in given provinces with types of cancer and $NO_x$.

| Province | $NO_2$ ICD/r | $NO_x$ ICD/r | Province | $NO_2$ ICD/r | $NO_x$ ICD/r |
|---|---|---|---|---|---|
| Lower Silesia (DS) | C96/0.65 $p = 0.006$ | C83/0.83 $p = 0.00007$ | Silesia (SL) | C33/0.61 $p = 0.012$ | C57/0.70 $p = 0.003$ |
| Kujawy-Pomerania (KP) | D07/0.55 $p = 0.027$ | D07/0.75 $p = 0.0008$ | Świętokrzyskie (SK) | C23/0.79 $p = 0.0003$ | C10/0.74 $p = 0.001$ |
| Lublin (LU) | C91/0.85 $p = 0.00003$ | C95/0.78 $p = 0.0004$ | Warmia-Masuria (WN) | C49/0.53 $p = 0.035$ | C24/0.80 $p = 0.0002$ |
| Lubusz (LB) | C57/0.72 $p = 0.002$ | D07/0.75 $p = 0.0008$ | Greater Poland (WP) | C92/0.65 $p = 0.006$ | C83/0.76 $p = 0.0006$ |
| Lodz (LD) | C00/0.68 $p = 0.004$ | C51/0.50 $p = 0.049$ | West Pomerania (ZP) | C18/0.77 $p = 0.0005$ | C50/0.67 $p = 0.005$ |
| Lesser Poland (MA) | C80/0.79 $p = 0.0003$ | C38/0.90 $P < 0.00001$ | | | |
| Masovia (MZ) | C17/0.72 $p = 0.002$ | C17/0.78 $p = 0.0004$ | | | |
| Opole (OP) | C14/0.78 $p = 0.0004$ | C26/0.62 $p = 0.010$ | | | |
| Podkarpackie (PK) | D06/0.67 $p = 0.005$ | C80/0.69 $p = 0.003$ | | | |
| Podlaskie (PD) | C33/0.87 $p = 0.00001$ | C88/0.61 $p = 0.012$ | | | |
| Pomerania (PM) | C31/0.66 $p = 0.005$ | C31/0.75 $p = 0.0008$ | | | |

r—Pearson correlation coefficient, ICD—International Classification of Diseases code.

The selection of provinces resulted from the reading presented in Table 2, in which bowel cancer (C17 and C18) was best correlated with $NO_x$ emissions.

In the last part of the analysis, we examined which contaminants have the greatest influence on cancer incidence of malignant neoplasm of the bronchus and lung (C34), and also of the small (C17) and large intestine (C18) by means of the random forest regression model. The random forest algorithm is an ensemble learning method for classification, regression, and other tasks that operate by constructing a multitude of decision trees at training time and outputting the class that is the mode of prediction (regression) of the individual trees. In the simplest terms, this method is based on the best fit of the function, in which the arguments are the measured amounts of gases and pollutants, and the result is the number of cases of a given type of cancer.

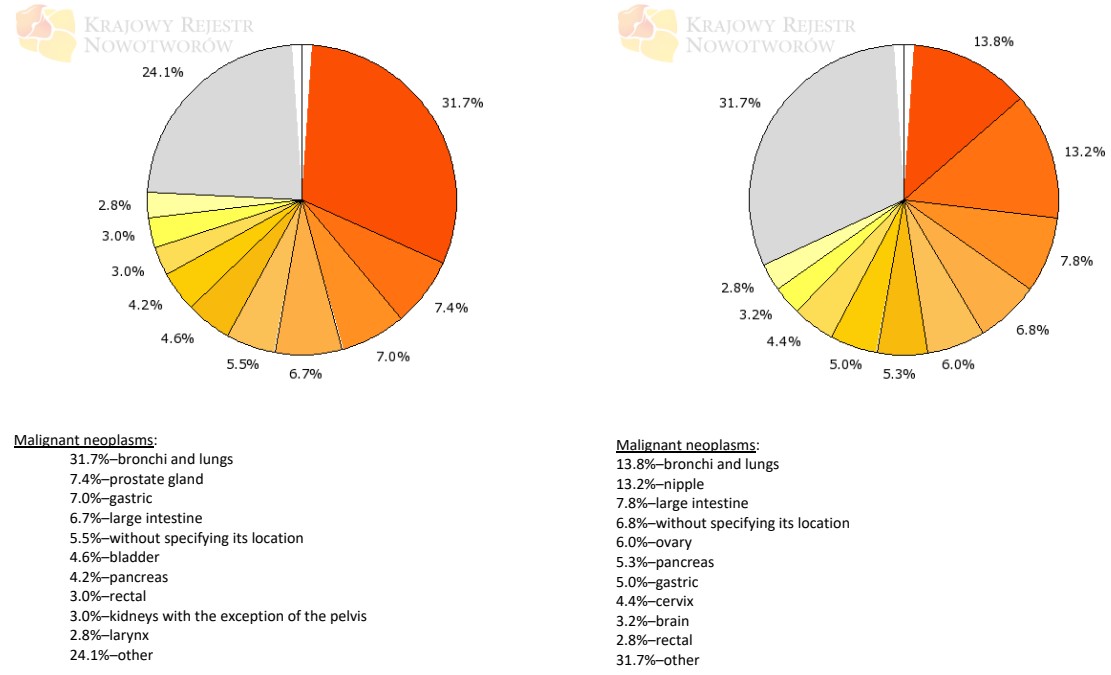

Malignant neoplasms:
  31.7%–bronchi and lungs
  7.4%–prostate gland
  7.0%–gastric
  6.7%–large intestine
  5.5%–without specifying its location
  4.6%–bladder
  4.2%–pancreas
  3.0%–rectal
  3.0%–kidneys with the exception of the pelvis
  2.8%–larynx
  24.1%–other

Malignant neoplasms:
  13.8%–bronchi and lungs
  13.2%–nipple
  7.8%–large intestine
  6.8%–without specifying its location
  6.0%–ovary
  5.3%–pancreas
  5.0%–gastric
  4.4%–cervix
  3.2%–brain
  2.8%–rectal
  31.7%–other

**Figure 4.** Cases of death as a result of malignant neoplasms in Poland in the years 1999–2015. On the left—men; on the right—women. Source: National Cancer Registry [10].

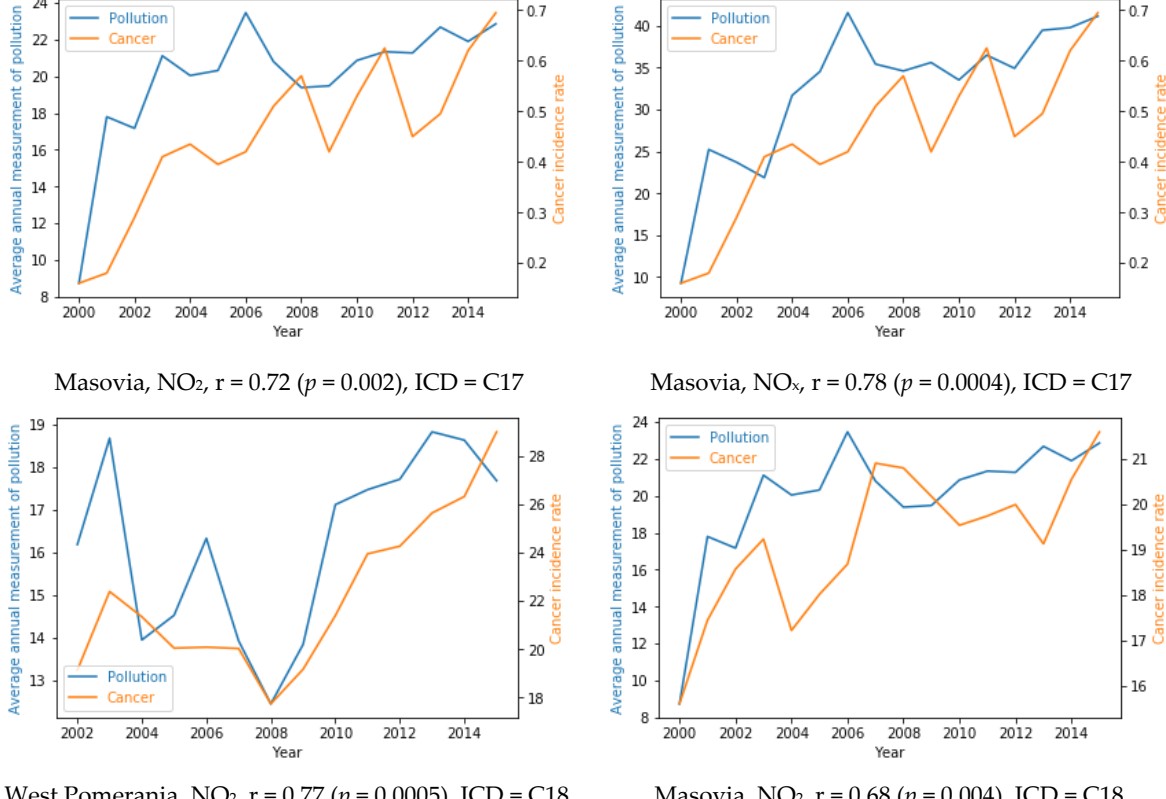

Masovia, NO$_2$, r = 0.72 ($p$ = 0.002), ICD = C17

Masovia, NO$_x$, r = 0.78 ($p$ = 0.0004), ICD = C17

West Pomerania, NO$_2$, r = 0.77 ($p$ = 0.0005), ICD = C18

Masovia, NO$_2$, r = 0.68 ($p$ = 0.004), ICD = C18

**Figure 5.** Level of nitrogen oxide emissions ($\mu g/m^3$) with the formation of bowel cancer, where r stands for the Pearson correlation coefficient and ICD is the International Classification of Diseases code.

In the case of lung cancer, it turned out that the basic contaminants affecting its formation are particles with a diameter of 2.5 μm or less (PM2.5). These contaminants may not be filtered by human organs, thus enabling toxic dust to penetrate into the lungs, bronchi, blood, and thus into the brain [21].

The results of the analysis are shown in Figure 6a–c.

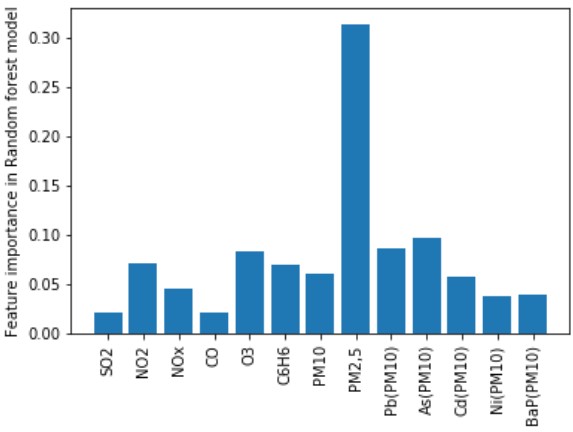

(**a**) Components of air pollution

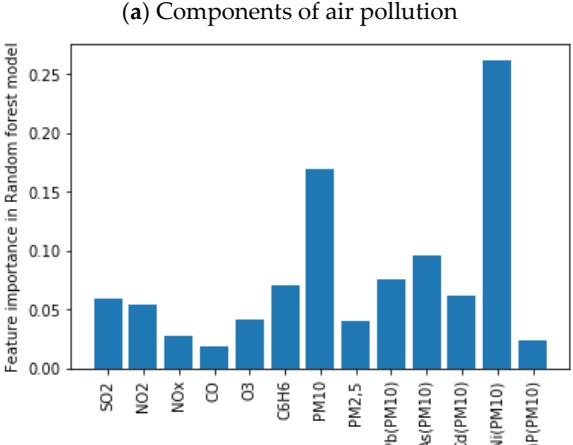

(**b**) Components of air pollution

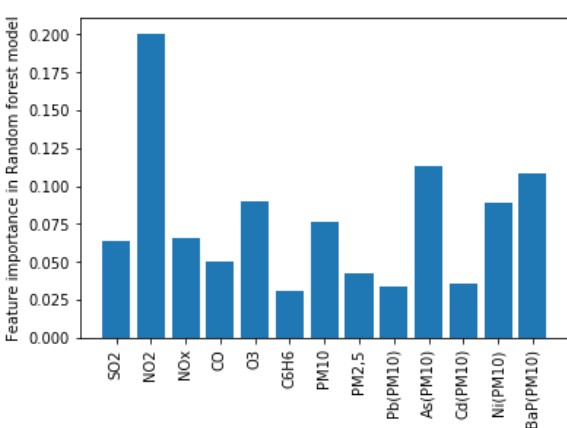

(**c**) Components of air pollution

**Figure 6.** The importance of parameters (types of pollutants) in the random forest regresson model when analyzing the causes of diseases (**a**) C34 (malignant neoplasm of bronchus and lung), (**b**) C17 (malignant neoplasm of the small intestine), and (**c**) C18 (malignant neoplasm of the large intestine).

The concept of feature importance refers to a class of techniques for assigning scores to input features to a predictive model that indicates the relative importance of each feature when making a prediction. It is quantitative parameter. As shown in Figure 6c, it can be seen that the three most important air pollutants that may affect the formation of malignant colon cancer are $NO_2$, As (PM10), and BaP (PM10).

Finally, the average accuracy of the random forest model was calculated, but the result was not high (only 20%). This means that air pollution is not the only factor in the formation of cancer in Poland. We may suggest that factors related to human nutrition, water quality, and smoking also need to be included.

## 4. Discussion

The first major observation in this study was a strong relationship between the level of PM 2.5 in the air and the incidence of lung cancer. Furthermore, we showed the effect of nitrogen oxides on the formation of tumors, and in particular the correlation between the presence of $NO_2$ in the air and the formation of colon cancer. Consequently, our data suggest that the level of $NO_2$ in the air and compounds present in the dust (arsenic, benzo(a)pyrene) occurring in the inhaled air may have a strong influence on the incidence of colorectal cancer.

Our results are in line with a very interesting correlation study performed in Japan, which examined the factors that could have caused the geographic variation observed in the lung and large intestinal cancer morbidity in that country. Lung cancer was highly correlated with industrialization-related factors such as localization of manufacturing industries, automobile traffic, and air pollution, whereas colon cancer was correlated with the population density of workers in the tertiary industries such as services, trade, and government. A multiple regression analysis could not detect any single factor with an exceptionally strong influence on either cancer [22].

An important problem when examining the factors contributing to the formation of specific cancer types is the proximity of residences to incinerators or hazardous waste disposal plants. The analysis of this problem was carried out in Spain and Italy. An increased cancer-related mortality in Spain was detected in the total population residing in the vicinity of these installations as a whole, and principally in the vicinity of incinerators and scrap metal/end-of-life vehicle handling facilities in particular. Special mention should be made of the results for tumors of the pleura, stomach, liver, kidney, ovary, lung, leukemia, colon/rectum, and bladder [23].

In the Italian analysis, no association between pollution exposure from the incinerators and all-cause and cause-specific mortality outcomes was observed in men, with the exception of colon cancer. However, exposure to the incinerators was associated with cancer mortality among women, in particular for stomach, colon, liver, and breast cancer. $NO_2$ levels as a proxy from other pollution sources (traffic in particular) did not exert an important confounding role [24].

The above may be of importance in relation to recent events in Poland. In the first half of 2018, nearly 70 landfill sites were burnt, and these fires may have similar effects as those mentioned in the abovementioned articles. As a result of the burning of rubber, plastic waste, and many kinds of chemical waste, poisonous and carcinogenic substances are created. Breathing polluted air increases the risk of cancer, which will pose a serious health issue in the near future.

## 5. Conclusions

Lung cancer is not the only cancerous threat related to air pollution. The latest research suggests that there are other cancers linked to air pollution. Nitrogen oxides have been shown to be the most strongly correlated type of gas with cancer statistics, and there are scientific grounds to attribute to it an influence on the development of serious illnesses. In Poland, the number of deaths attributed to long-term exposure to $NO_2$ is estimated at 1600 annually. It is worth mentioning that nitrogen oxides also harm us indirectly. They are precursors of carcinogenic compounds formed in soils that can

penetrate into food. In this case, their impact on the incidence of chronic diseases and, as a consequence, on mortality is very difficult to estimate [25].

Our study showed that:

1. There are strong correlations in given provinces with the type of cancer.
2. Based on the analysis, it was found that the formation of C17 and C18 disease (colorectal cancer) is the most strongly correlated with the emission of nitrogen oxides in the Masovia province and in the West Pomeranian region.
3. Analysis of the influence of the type of air pollutants on the formation of selected types of cancer showed that:

   - for lung cancer, the release of PM2.5 pollution plays the most important role,
   - the most important issue in colorectal cancer is the emission of nitrogen dioxide.

4. Emissions of nitrogen dioxide, as well as arsenic and benzoalfapiren compounds found in suspended dust, have an effect on the development of large intestinal diseases (C18).

Our study points to the need for in-depth air pollution data collection, such as with sensors and drones, to allow for further characterization and exposure assessment. Due to the non-uniform location of measurement stations, more accurate measurements of hazardous substances could be then carried out, especially when using a swarm of drones [26,27].

**Author Contributions:** N.T.—Writing—original draft, J.F.—Writing—review & editing, P.N.—Project administration, P.T.—Software. All authors have read and agreed to the published version of the manuscript.

**Funding:** This research received no external funding.

**Acknowledgments:** We would like to thank the contractors of the project "Establishing the first IT platform in Poland to exchange knowledge about the threat of malignant neoplasms in Poland" carried out in 2010–2013 by the National Cancer Registry, located in the Oncology Center, Maria Skłodowska-Curie Institute in Warsaw, for creating a database. Supported by the grant from the Medical University of Lodz (#503/1-156-04/503-11-001-19-00 to J.F.).

**Conflicts of Interest:** The authors declare no conflict of interest.

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
