# Peer review of "Air Pollution Associates with Cancer Incidences in Poland"

_applsci, doi:10.3390/app10217489_

Round 1
Reviewer 1 Report
This is a well-written and informative manuscript for readers to understand the significant correlations between respective pollutant mentioned and the cancer type.
Reviewer lists some points below for Authors' attentions.
- In the "Methods" section, authors can provide more details about how the Python codes were used to estimate the Pearson correlation coefficient (PCC) and the Random Forest regression algorithm results, i.e., with some simple examples to show the readers how the statistical analysis was done, so the readers have more ideas on PCC and Random Decision Forests algorithm.
- Reviewer suggests the Python codes used for this manuscript to be posted (with link information in the manuscript) and accessible by the readers so researchers can use those Python codes to do similar study and analysis if possible.
- On Line 261, "coroteral" (cancer) should be a typo, reviewer guess the authors meant "colorectal" (cancer) here.
Author Response
Dear Reviewer,
thank you very much for your valuable tips for improving the article.
- In the "Methods" section, authors can provide more details about how the Python codes were used to estimate the Pearson correlation coefficient (PCC) and the Random Forest regression algorithm results, i.e., with some simple examples to show the readers how the statistical analysis was done, so the readers have more ideas on PCC and Random Decision Forests algorithm.
More details are provided on how Python codes were used to estimate the Pearson correlation coefficient (PCC) and the results of the Random Forest regression algorithm, and what packages and libraries were used.
- Reviewer suggests the Python codes used for this manuscript to be posted (with link information in the manuscript) and accessible by the readers so researchers can use those Python codes to do similar study and analysis if possible.
The source code used for analysis is posted on repository (https://github.com/ntusnio/CV/blob/master/Projekt%20rak.ipynb) and this information was added.
- On Line 261, "coroteral" (cancer) should be a typo, reviewer guess the authors meant "colorectal" (cancer) here.
This clerical error has been corrected.
Reviewer 2 Report
Review
Overall comments.
The current paper aims to evaluate the relationship between selected air pollutants and cancers in Polish population. Data on air quality measurements were obtained from the Chief Inspectorate for Environmental Protection site, a data base with results from 264 stations located in Poland for 13 pollutants from 2000 to 2016 and retrospective data from Polish Cancer registry between 1999 and 2016. The investigators correlated the trends in air pollutant levels with cancer incidence rates overall and by selected pollutant and cancers to find positive correlation between air pollution with incidence of selected types of cancer. While past evidence from various studies in the US and Europe have suggested possible association between air pollutants and cancer, the present paper has several methodological concerns which precludes it from adding any robust evidence to support the hypothesis.
Major comments.
While the authors have discussed evidence from previous studies in their introduction, it would be helpful to the readers to present comparative statistics such as measures of association (relative risks, correlation) to provide some understanding of the strength of association between the air pollution and the cancers.
A table summarizing the past literature (only relevant studies) including types of pollutants studied, levels of pollutants, study designs and target population as compared to the data in the current study would provide evidence of external validity of the study.
A major issue with the design of the study is the temporality of exposure and outcome. Most cancers, except for hematological cancers and childhood cancers, have long induction periods >10 years. Based on the graphs, the authors have presented correlations between annual average pollutant levels and annual incidence rates of the same year across Poland. In fact, the start of exposure assessment is from 2000 and the outcome is 1999, indicating that exposure follows the outcome or at least is concurrent with outcome. This does not support the biological plausibility of long latency period of exposure needed to cause carcinogenesis. The high correlation observed between pollution levels and incidence rates could be a function of confounding factors resulting from trends in population demographics and should be considered in the analyses.
The authors may want to consider using time-series analyses with long lag periods ( 5, 10, 15 or 20 years) to model exposure to address this issue.
It is not clear if age-adjusted incidence rates were used.
Author Response
Dear Reviewer,
thank you very much for your valuable tips for improving the article.
Below, we have allowed ourselves to post replies to comments and information about what has been done.
While the authors have discussed evidence from previous studies in their introduction, it would be helpful to the readers to present comparative statistics such as measures of association (relative risks, correlation) to provide some understanding of the strength of association between the air pollution and the cancers.
The associations mentioned by the reviewer are not only in Intro, but also in the discussion from line 231.
The authors found a very similar article for Taiwan (all cancer cases 2012-2016) and tried to cite it:
https://bmcpublichealth.biomedcentral.com/articles/10.1186/s12889-019-7849-z
A table summarizing the past literature (only relevant studies) including types of pollutants studied, levels of pollutants, study designs and target population as compared to the data in the current study would provide evidence of external validity of the study.
We made a suggested table.
A major issue with the design of the study is the temporality of exposure and outcome. Most cancers, except for hematological cancers and childhood cancers, have long induction periods >10 years. Based on the graphs, the authors have presented correlations between annual average pollutant levels and annual incidence rates of the same year across Poland. In fact, the start of exposure assessment is from 2000 and the outcome is 1999, indicating that exposure follows the outcome or at least is concurrent with outcome. This does not support the biological plausibility of long latency period of exposure needed to cause carcinogenesis. The high correlation observed between pollution levels and incidence rates could be a function of confounding factors resulting from trends in population demographics and should be considered in the analyses.
The authors may want to consider using time-series analyses with long lag periods (5, 10, 15 or 20 years) to model exposure to address this issue.
The authors considered using time-series analyses with longer lag periods when doing the calculations, but were unsure of the choice of the lag period and stayed with zero lag. In simplified terms, it has been assumed that there will be cases of neoplasms in the area exposed to air contamination. The data that were available also did not allow for taking into account human migration and the fact that they could heal themselves in a different region than they inhaled the air.
It is not clear if age-adjusted incidence rates were used.
No age or gender-adjusted incidence rates were used.
Reviewer 3 Report
The authors of this manuscript present interesting results on the relationship between air pollution and cancer. While there is value in their work since such data throughout the whole of Poland have not be extensively studied, there are several improvements that can be made to the current write up.
Line 39: ...exposure to ionizing radiation and environmental pollution are listed as number 1 or number two? what is the importance? I ask this because the next sentence introduces air pollution for the first time. Hence in line 39 if authors are referring to ambient air pollution for instance, then please do so without beating about the bush (ie referring to environmental pollution).
I would also define "PM" for the first time in the introduction.
Lines 45 to 48, please include reputable references for the composition of air pollutants.
Line 58: The opening sentence is confusing...the authors provide citation #4 which is a Canadian study, yet they are discussing studies in Spain and the US with no citations. Please revise and briefly describe and cite each study, or else delete that section.
Delete "in this country" in this country makes it sound like in Poland, but authors were referring to the US
Line 71: Studies carried out at the National Statistics Institute in Spain have allowed to designate the designation of industries that contribute...
Line 75: whether the proximity of [such] industries
Lines 77-80: Are there any references to find further information
Line 84: environment can also [be] made with...
line 96 how and why were these types of cancer selected, can authors briefly explain
line 109 I am not familiar with "PM10 and PM2.5 dust" perhaps authors can state "ambient PM" since they correctly state in the beginning that it is not just dust, that is found in PM
lines 110-111 not sure what positions are...do the authors mean locations?
line 155 and methods section, why not truncate air pollution data at2015, just like cancer data? It seems from results that data are indeed up to 2015, so why did the authors state that they had data till 2016? There is no need for the extra year of data unless some prediction models are incorporated. Did the authors want to do some prediction modelling with this data set? If so where is that presented?
page 6. Please check the map as some provinces are cut off
Please reword to something like this in the ff places:
Line 214: the first major observation in this study was a strong relationship between PM254 and lung cancer incidence
line 260: for lung cancer PM2.5 plays an important role
Line 264: Our study points to the need for in depth air pollution data collection, such as with sensors and drones to allow for further characterization and exposure assessment...
Author Response
Dear Reviewer,
thank you very much for your valuable tips for improving the article.
Line 39: ...exposure to ionizing radiation and environmental pollution are listed as number 1 or number two? what is the importance? I ask this because the next sentence introduces air pollution for the first time. Hence in line 39 if authors are referring to ambient air pollution for instance, then please do so without beating about the bush (ie referring to environmental pollution).
We agree with the Reviewer, therefore the sentence has been rephrased.
I would also define "PM" for the first time in the introduction.
In the introduction we added that PM stands for particulate matter.
Lines 45 to 48, please include reputable references for the composition of air pollutants.
The reference (book) about the composition of air pollutants was included.
Line 58: The opening sentence is confusing...the authors provide citation #4 which is a Canadian study, yet they are discussing studies in Spain and the US with no citations. Please revise and briefly describe and cite each study, or else delete that section.
We agree with the Reviewer, therefore the sentence has been rephrased and proper reference use double-checked.
Delete "in this country" in this country makes it sound like in Poland, but authors were referring to the US
Words "in this country" were deleted.
Line 71: Studies carried out at the National Statistics Institute in Spain have allowed to designate the designation of industries that contribute...
Words “the designation of“ were added.
Line 75: whether the proximity of [such] industries
Word “such” was added.
Lines 77-80: Are there any references to find further information
This is a World Health Organization (WHO) ranking of the EU cities with the most polluted air.
https://airly.org/pl/zanieczyszczone-powietrze-w-polsce-sprawdz-regiony-szczegolnie-niebezpieczne/
https://notesfrompoland.com/2020/01/17/polish-city-registers-second-worst-air-pollution-in-the-world-as-smog-descends-on-poland/
Line 84: environment can also [be] made with...
Word “be” was added.
line 96 how and why were these types of cancer selected, can authors briefly explain
During the analysis of the primary results, data appeared that allowed to focus on these types of cancers. In Poland, the most common neoplasms in men are lung neoplasms, accounting for about 1/5 of cancer incidence. These are followed by prostate cancer (13%), colorectal cancer (12%) and bladder cancer (7%). The most common cancer among women is breast cancer, accounting for over 1/5 of cancer cases, colorectal cancer is the second most common cancer in women (10%), lung cancer is third (9%).
line 109 I am not familiar with "PM10 and PM2.5 dust" perhaps authors can state "ambient PM" since they correctly state in the beginning that it is not just dust, that is found in PM
The sentence has been modified by replacing "dust" with "emission".
lines 110-111 not sure what positions are...do the authors mean locations?
Words “positions” were replaced by “locations”.
line 155 and methods section, why not truncate air pollution data at 2015, just like cancer data? It seems from results that data are indeed up to 2015, so why did the authors state that they had data till 2016? There is no need for the extra year of data unless some prediction models are incorporated. Did the authors want to do some prediction modelling with this data set? If so where is that presented?
With the available pollution data for 2000-2016 and cancer cases for 1999-2015, the analysis could be performed only for the overlap years (2000-2015).
page 6. Please check the map as some provinces are cut off
The map is fully visible when the table is on a new page.
Please reword to something like this in the ff places:
line 214: the first major observation in this study was a strong relationship between PM254 and lung cancer incidence
line 260: for lung cancer PM2.5 plays an important role
line 264: Our study points to the need for in depth air pollution data collection, such as with sensors and drones to allow for further characterization and exposure assessment...
The phrases have been reworded accordingly.
Round 2
Reviewer 2 Report
Given the scope of this paper and analyses plan presented in the manuscript, I firmly believe that the manuscript has to undergo substantial changes before being published. While the authors have cited a previous manuscript published with similar approach, it does not necessarily support the approach taken by the researchers in this paper.
The authors have presented correlations in the study and trends in levels of air pollution over years correlated with cancer incidence over time. Since only correlations have been used to compare exposure and outcomes, no conclusions can be drawn regarding impact or risk of air pollutants on cancer.
The authors do not address the issue of latency (critical for any cancer-related studies).
Most of the evidence supporting air pollutants and cancers come from occupational exposure or prospective studies with >20 years of follow up (1,2). Exposure to toxicants in air pollution are at low levels and would require long duration of exposure for development of cancer (3). Without taking into account the latency or lagged exposure, presenting correlations over time is uninformative. I am thus not convinced that the manuscript should be published in its current form.
References:
- Kim HB, Shim JY, Park B, Lee YJ. Long-Term Exposure to Air Pollutants and Cancer Mortality: A Meta-Analysis of Cohort Studies. Int J Environ Res Public Health. 2018;15(11):2608. Published 2018 Nov 21. doi:10.3390/ijerph15112608
- White AJ, Keller JP, Zhao S, Carroll R, Kaufman JD, Sandler DP. Air Pollution, Clustering of Particulate Matter Components, and Breast Cancer in the Sister Study: A U.S.-Wide Cohort. Environ Health Perspect. 2019;127(10):107002. doi:10.1289/EHP5131
- Zhonghuan Xia, Xiaoli Duan, Shu Tao, Weixun Qiu, Di Liu, Yilong Wang, Siye Wei, Bin Wang, Qiujing Jiang, Bin Lu, Yunxue Song, Xinxin Hu, Pollution level, inhalation exposure and lung cancer risk of ambient atmospheric polycyclic aromatic hydrocarbons (PAHs) in Taiyuan, China. Environmental Pollution,Volume 173,2013,Pages 150-156.,
Author Response
Dear Reviewer,
Thank you once again for your valuable comments and literature items.
Our studies were designed without taking into account the latency as in some other cancer research. We also realize that each tumor has a different period of carcinogenesis.
This approach to the subject is very laborious and requires new calculations, for which we will not have enough time due to publishing deadlines.
We can make a commitment that such analysis will be performed and the results published in the next approach.
We would like to note that this article has already been uploaded to Applied Sciences twice, the first time we got 2 positive reviews, and the second time we got 2 positive reviews, so it can be assumed that we have 4 positive reviews in total, but we need 3 in one go.
So please agree to our request to publish the article in its current form.
Best regards,
Authors
This manuscript is a resubmission of an earlier submission. The following is a list of the peer review reports and author responses from that submission.
Round 1
Reviewer 1 Report
Dear Authors,
The authors state that ‘The aim of the study was to indicate which types of cancer are characterized by the highest correlation with the emission of selected types of air pollutants’. However, throughout the paper, pollutant emissions have not been studied or quantified. What are the pollutant emissions across Polish territory? Probably the authors wanted to say that they intend to study the correlation of the occurrences of cancer and the exposure of the population to air pollutants.
The relationship between the occurrence of diseases and exposure to pollutants should not be made in such a simplistic way. There are a number of factors that strongly influence the occurrence of disease, such as habits of life, food, lifestyle ...
The article is very poorly written, poorly formatted, poor analysis of results and poor conclusions were made. I do not recommend publishing this article.
Some changes that should be made:
- Change ‘ powietrze.gios.gov.pl’ by the name of the site (line 20)
- Change ‘Source:http://onkologia.org.pl/raporty/#wykres_piramida_wieku’ by the name of the site (line 99) and insert the the link in the bibliography
- The numbering of the references must be corrected
- Insert the meaning of ICD-10 (line107)
- Incorrect formatting (Table 1 and Figure 4…)
- What is the meaning of the line 155?
- The text between lines 189-199 should be moved to the introduction chapter
…
Author Response
Dear Reviewer,
We would like to thank you for your critical comments regarding our article.
The calculations were made as part of the final work on the "Data science" course and consisted of checking the correlation of data in 2 databases.
In Poland, in the Pomeranian Province, the incidence of malignant neoplasms is at one of the highest levels in Poland. The lack of a clearly defined reason is conducive to the development of conspiracy theories, e.g. on radioactive waste in the Baltic. The solution can be trivial – in Pomerania, people live longer than in other provinces, and a longer age favors the development of cancer.
In Poland, tobacco addiction seems to be an important factor contributing to high mortality, but the authors did not reach data regarding smokers who had cancer.
All comments have been entered (minor corrections can be made in the second iteration).
Authors
Reviewer 2 Report
Please see attached file.

Author Response
Dear Reviewer,
We would like to thank you for your critical comments regarding our article.
We have addressed all comments.
Authors
- Line 13 clarified
- Line 17 clarified
- Lines 24–25 changes made
- Line 28 change made
- Keywords changed
- Line 48 change made
- Line 92 change made
- Line 98 The y-axis has been described (age). Chart generated from the site http://onkologia.org.pl/raporty/#wykres_piramida_wieku. When choosing the years range, patients are added up. Calculations from several years may concern the same patient. There are no decimal points. The scale for statistics from one year is 2 million on the left and 2 million on the right.
- Lines 100–108 information has been completed
- Table 1 done
- Lines 120-127 done
- Figure 2 The y-axis shows the time-averaged measurement of pollution (on an annual basis). This should be one point for each year, but this visualization of the data seems more visually appealing (the data is actually not continuous). The unit of measurement for dusts and gases is µg/m3. Exceptionally for CO it is mg/m3. The p-value for each Pearson correlation coefficient was added.
- Line 136 done
- Figure 3 Not applicable, these are bar charts.
- Line 143 entered
- Table 2 There is not much space in the table to expand the ICD, but a map has been inserted.
- Lines 149-150 corrected
- Figure 4 corrected
- Line 157 entered
- Line 163 Rather, it is about the dynamics of disease growth. Nothing changed in the manuscript. We want to clarify that it's about dynamics. And that's why this is our main focus - even if in some provinces we have higher rates for selected diseases.
- Figure 5 The average annual measurement of pollution means the value of pollution averaged over a period of 1 year for a given province. One point falls for each year, but the points are combined to increase the attractiveness of the chart (in fact, the data is not continuous). The p-value for each Pearson correlation coefficient was added.
- Line 167 done
- Lines 169-173Last sentence can be deleted, we tried to explain the method.
- Line 177 Citation provided. The sentence has been rephrased.
- Figure 6 The axis descriptions have changed. Feature importance was explained.
- Lines 185-186 added that the accuracy score was not high
- Line 200 changed
- Lines 201-204 changed
- Lines 224–228 sentence changed
Reviewer 3 Report
I found this article to be interesting, but not well written. I think the authors have tried to put their findings into context and many of my suggestions are for clarifications. The major points they should consider are the following:- It's really unclear to me the scientific hypothesis that the authors take into consideration. This is an important point before we can start to reason about the proposed method seriously.
- The English in the present manuscript is hard to understand and requires major improvement to meet the publication quality.
- There are other concerns listed as below:
a. Line 133, The symbol "r" is the most commonly used letter for Pearson's correlation when measured in a sample, not "PCC" used in this manuscript. The author also needs to provide "p-value" to tell the significant levels. The question is, what's the sample size when you performed Pearson's correlation?
b. Line 103-104, What the chemical formula "C6H6" stands for? NOx is a common designation of nitrogen oxides NO and NO2. Is that any reason you listed NO2 as the measured substance separately?
Author Response
Dear Reviewer,
We would like to thank you for your critical comments regarding our article.
We have addressed all comments.
Authors
- The Pearson correlation coefficient symbol has been changed and the p-value has been calculated. The sample size for years 2000-2015 was N = 16.
- C6H6 is a symbol of benzene. It is one of the substances whose concentration is measured as part of the air quality test in Poland. NOx means all nitrogen oxides (a mixture of nitrogen oxides with an undefined composition). NOx and NO2 were available in the measurement database and both were used.
Round 2
Reviewer 1 Report
Please, see below.
Reviewer 2 Report
The authors have made corrections to the manuscript but they are insufficient to warrant consideration for publication.
Reviewer 3 Report
Dear Authors,
Thanks for addressing all my concerns from the reviews. Your revised version of the manuscript appears to be good.
Thanks!